# Bayesian Disease Progression Models that Capture Health Disparities

## Abstract

Disease progression models, in which a patient's latent severity is modeled as progressing over time and producing observed symptoms, have developed great potential to help with disease detection, prediction, and drug development. However, a significant limitation of existing models is that they do not typically account for healthcare disparities that can bias the observed data. We draw attention to three key disparities: certain patient populations may (1) start receiving care only when their disease is more severe, (2) experience faster disease progression even while receiving care, or (3) receive care less frequently conditional on disease severity. To address this, we develop an interpretable Bayesian disease progression model that captures these three disparities. We show theoretically and empirically that our model correctly estimates disparities and severity from observed data, and that failing to account for these disparities produces biased estimates of severity.

## 1 Introduction

Using observed data to model the progression of a latent variable over time is useful for making predictions in many settings. Models of infrastucture deterioration use physical observations and inspection results to model a system's overall health changing over time [1]; models of human aging use a person's observed physical and biological characteristics to learn the progression of their underlying "biological age" [2]; and disease progression models, the setting we focus on in this paper, use observed symptoms to learn a patient's evolving latent disease severity [3]. Disease progression models provide insight on both individual-level disease trajectories and general representations of disease dynamics. Accurately modeling disease progression offers great promise in enabling healthcare providers to better personalize care and predict a patient's disease trajectory, detect diseases at earlier stages, and study interventions such as drug development [4, 5].

In order for the benefits of these models to apply to all patients equitably, it is crucial that they make accurate predictions for all populations of patients. However, disease progression models have typically failed to account for systemic disparities in the healthcare process. Disparities have been shown to exist along many demographic features including socioeconomic status [6, 7], proximity to care [8, 9], and race [10] — intuitively, we expect that models not accounting for these disparities will make predictions that are consistently inaccurate for some patient groups. In this paper, we define three main axes along which we observe and analyze disparities:

1. Certain patient groups may start receiving care only when their disease is more severe (leaving more of their disease trajectory unobserved).

2. Certain patient groups may experience faster disease progression even while receiving care (indicating consistent differences in the efficacy or quality of treatment).

Submitted to Workshop on Bayesian Decision-making and Uncertainty, 38th Conference on Neural Information Processing Systems (BDU at NeurIPS 2024). Do not distribute.

3. Certain patient groups may receive care less frequently conditional on disease severity (decreasing the frequency with which they are observed in the data).

As such, our key contributions are: (1) we propose an interpretable Bayesian model that learns disease progression while accounting for disparities along all three of these axes, (2) we show theoretically and empirically that failing to account for any of these disparities will lead to biased severity estimates, and (3) we outline the beginning of a heart failure case study. We anticipate that the results from this case study, which we are working on in close collaboration with the New York-Presbyterian hospital system, will have two main applications: descriptions of healthcare disparities across demographic groups can help to target future interventions, and validating the model in a real healthcare setting will demonstrate that it is possible to make predictions without bias from these disparities.

## 2   Related Work

**Disease progression modeling.**   Disease progression models have been developed for many chronic diseases, including Parkinson's disease [3], Alzheimer's disease [11], diabetes [12], and cancer [13]. A key feature of the progression models we consider is that a latent severity $Z_t$ progresses over time and gives rise to the observed symptoms $X_t$. Models in this family include variants of hidden markov models (HMM) [14, 15, 16, 17, 18] and recurrent neural networks (RNN) [19, 20, 21, 22, 23, 24, 25].

**Healthcare disparities.**   Disparities have been documented in many parts of the healthcare process. Factors such as distance from hospitals [8, 9], distrust of the healthcare system [26], or lack of insurance [27] can result in underutilization of health services. Biases in the judgements of healthcare providers can lead to minority groups receiving later screening [28], fewer referrals [29], or generally worse care [30]. And issues such as limited health literacy or trust in healthcare can create disparities in follow-through for appointments or effectiveness of at-home care [31, 32].

These disparities have been shown to emerge along the three axes that we identify: (1) how severe a patient's disease gets before they start to receive care [33, 34, 35]; (2) how quickly their latent severity $Z_t$ progresses even while receiving care [36, 37]; and (3) how likely they are to visit a clinician at a given disease severity level [38]. Despite thorough literature showing the existence of these disparities and their impact on healthcare, disease progression models have not (to the best of our knowledge) accounted for disparities when making predictions.

## 3   Model

We build on a standard setup for disease progression modeling, in which each patient $i$ has an underlying latent disease severity $Z_t^{(i)}$ that progresses over time and gives rise to a set of observed features $X_t^{(i)}$ [39, 40]. For notational convenience, we will omit the $(i)$ superscript from here on.

We characterize a patient's severity $Z_t \in \mathbb{R}$ at timestep $t$ by their *initial severity* $Z_0$ at their first observation (which we denote as $t = 0$) and their *rate of progression* $R$ after that point:

$$Z_t = Z_0 + R \cdot t$$

While we expect our approach to extend naturally to non-linear models of progression, estimating the slope of a potentially non-linear progression still provides valuable insight on a patient's general disease trajectory relative to others. The assumption of linear progression over time to capture long-term disease trajectory is a common approach in existing models [11, 2].

Whether a patient actually visits a healthcare provider at time $t$ is captured by an observed binary indicator $D_t \in \{0, 1\}$. If a patient does visit at time $t$, we will observe some recorded set of disease-relevant features $X_t \in \mathbb{R}^d$ (e.g., lab results, imaging, and symptoms). At any given timestep, a clinician will not necessarily observe or record all features — we model the features that *are* observed as a noisy function of latent severity $Z_t$:

$$X_t = f(Z_t) + \epsilon_t$$

where diagonal covariance matrix $\sigma_\epsilon \in \mathbb{R}^{d \times d}$ parameterizes feature-specific noise $\epsilon_t \sim N(0, \sigma_\epsilon)$ (accounting for both measurement error and variation in how the patient's physical state can fluctuate day-to-day). We specifically instantiate $f$ as a linear function $f(Z_t) = F \cdot Z_t + F_{int}$, where

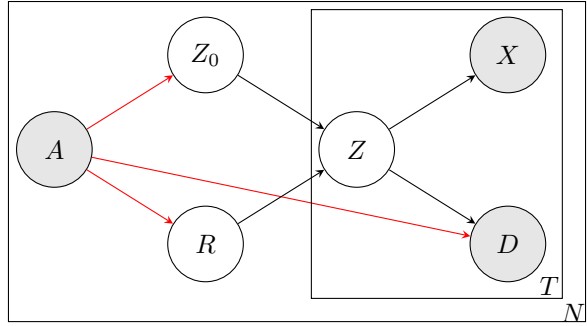

Figure 1: Plate diagram of generative model, capturing $N$ patients over $T$ timesteps. Shaded nodes indicate observed features, and red arrows indicate dependencies capturing health disparities.

$F_{int} \in \mathbb{R}^d$ is a feature-specific intercept and $F \in \mathbb{R}^d$ has its first element constrained to be positive for identifiability; we leave extending this to non-linear functions for future work.

**Capturing disparities.** We next specify a demographic feature vector $A$ for each patient. $A$ can capture multiple social determinants of health (each element of $A$ can encode any continuous or categorical feature), but for simplicity in exposition, we assume $A$ encodes a single categorical label (e.g., a patient's race group). By modeling dependence between $A$ and other aspects of the model, depicted in Figure 1, we can capture health disparities along three interpretable axes; as we discuss in §2, the existence of these disparities has been well-documented in past studies:

1. **Underserved patients may start receiving care only when their disease is more severe.** We capture this by learning group-specific distributions of $Z_0$, a patient's disease severity at first visit. We pin $Z_0$ for one group ($A = a_0$) to be drawn from a unit normal distribution (as is standard because it fixes the scale of $Z_t$). For other groups $A = a$, $Z_0 \sim N\left(\mu_{Z_0}^{(a)}, \sigma_{Z_0}^{(a)}\right)$, where $\mu_{Z_0}^{(a)}$ and $\sigma_{Z_0}^{(a)}$ are learned group-specific parameters for group $a$.

2. **Underserved patients may experience faster disease progression even while receiving care**. This we capture by learning group-specific distributions of progression rate $R \sim N\left(\mu_R^{(a)}, \sigma_R^{(a)}\right)$, where $\mu_R^{(a)}$ and $\sigma_R^{(a)}$ are learned group-specific parameters for group $a$.

3. **Underserved patients may receive care frequently conditional on disease severity.** This we capture by modeling patient visits as generated by an inhomogeneous Poisson process parameterized by a non-negative, time-varying rate parameter $\lambda_t$ that depends on both $Z_t$ and $A$ for all groups $a$: $\log(\lambda_t) = \beta_0 + (\beta_Z \cdot Z_t) + \beta_A^{(a)}$, where $\beta_Z$ and $\beta_0$ are learned parameters for the entire population and $\beta_A^{(a)}$ is a learned group-specific parameter for group $a$. We pin $\beta_A^{(a_0)}$ at 0 as a reference for all other groups.

Overall, our model parameters (on which we place weakly informative priors) are $F, F_{int}, \sigma_\epsilon, \{\mu_{Z_0}^{(a)}\}, \{\sigma_{Z_0}^{(a)}\}, \{\mu_R^{(a)}\}, \{\sigma_R^{(a)}\}, \beta_0, \beta_Z$, and $\{\beta_A^{(a)}\}$ for all demographic groups $a$. We learn these values from our observed data $X_t, D_t$, and $A$. Figure 1 summarizes the data generating process.

# 4 Theoretical analysis

## 4.1 Identifiability

As we show in §A.1, our model is identifiable, meaning different sets of parameters yield different observed data distributions [41, 42]:

**Theorem 4.1.** *All parameters of the model are identified by $P(X_t, D_t \mid A)$.*

We confirm our theoretical identifiability results experimentally in §5, showing that the model does indeed recover the true parameters in synthetic data.

## 4.2 Bias in models that do not account for disparities

Next we show that disease progression models will produce biased estimates of severity if they fail to account for any of the three disparity types we capture. We use the strict Monotone Likelihood Ratio Property (MLRP) to characterize the existence of disparities between two populations [43]. Our results apply to any setting in which data is generated according to the relationships depicted in Figure 1 and disparities exist, not relying on the parametric assumptions of our implemented model.

First, we prove that any model failing to account for disparity 1 will produce biased severity estimates:

**Theorem 4.2.** *A model that does not take into account demographic disparities in initial disease severity $Z_0$ will underestimate the disease severity of groups with higher values of initial severity and overestimate that of groups with lower values of initial severity.*

That is (for the underestimation case), if $P(Z_0 = z_0 \mid A = a)$ strictly MLRPs $P(Z_0 = z_0)$ for some group $a$, then $\mathbb{E}[Z_t \mid X_t = x_t] < \mathbb{E}[Z_t \mid X_t = x_t, A = a]$. A full proof is provided in §B.1. We then prove that failing to account for disparity 2 or disparity 3 will also lead to biased estimates of severity (full proofs in §B.2 and §B.3, respectively):

**Theorem 4.3.** *A model that does not take into account demographic disparities in rate of progression $R$ will underestimate the disease severity of groups with higher progression rates and overestimate that of groups with lower progression rates.*

**Theorem 4.4.** *A model that does not take into account demographic disparities in visit frequency $\lambda_t$ will underestimate the disease severity of groups with lower visit frequency and overestimate that of groups with higher visit frequency.*

# 5 Synthetic experiments

We implement our model in Stan, a Bayesian inference package [44], to validate our theoretical results in simulations with synthetic data.

## 5.1 Identifiability

We first verify Theorem 4.1 in simulations, showing our model can accurately recover the true data-generating parameters for synthetic data. Across 50 runs, we find high correlation between the true parameters and the posterior mean estimates (mean Pearson's $r$ 0.98 across all parameters; median 0.98), and good calibration (mean linear regression slope 0.97; median 0.98). We provide scatterplots of all parameter recovery in Appendix C.

## 5.2 Bias in models that do not account for disparities

We now verify in simulation that failing to account for disparities can lead to biased severity estimates. We generate simulated data for two groups, $A = 0$ and $A = 1$, where group 1 is underserved with respect to each of the three disparities we capture (i.e., $\mu_{Z_0}^{(1)} > \mu_{Z_0}^{(0)}$, $\mu_R^{(1)} > \mu_R^{(0)}$, and $\beta_A^{(0)} > \beta_A^{(1)}$). We then fit our main model, which accounts for all disparities, alongside three models that each fail to account for one of the disparities, on the same set of data to compare their recovery of individual patient severity values. As seen in Figure 2, the models that do not account for disparities all underestimate severity for the underserved group 1 and overestimates severity for the other group — these simulations empirically support Theorems 4.2, 4.3, and 4.4. While our main model achieves average error (mean inferred estimate minus mean true value for a single run) $-0.004$ and $-0.02$ for groups 0 and 1 respectively, the other models have error 1.03, 0.01, and 0.42 for group 0 (all overestimated) and error $-0.78$, $-0.24$, and $-0.88$ for group 1 (all underestimated).

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

# A    Identifiability Proofs

## A.1    Proof of Theorem 4.1

*Proof.* We want to show that each set of parameter assignments leads to a different distribution over the observed data. To do this, we divide our argument into four lemmas:

**Lemma A.1.** *Parameters $F, F_{int}, \sigma_\epsilon$ are identified by $P(X_t \mid A = a_0)$.*

> *Proof.* would probably cut the restatement of model definitions here and throughout the proofs.First we restate relevant details of the generative model for group $a_0$:
>
> $$Z_0 \sim N(0, 1)$$
>
> $$Z_t = Z_0 + R \cdot t$$
>
> $$X_t = F \cdot Z_t + F_{int} + \epsilon_t, \text{ where } \epsilon_t \sim N(0, \sigma_\epsilon) \tag{1}$$
>
> We first note that at $t = 0$ we have $Z_t = Z_0$ and thus $Z_t \sim N(0, 1)$. Then equation (1) captures a factor analysis modelwe should cite the source of the expression with factor loading matrix $F$ and diagonal covariance matrix $\sigma_\epsilon$. So at $t = 0$, we have for group $a_0$ that
>
> $$X_0 \sim N(F_{int}, FF^T + \sigma_\epsilon).$$
>
> my guess is that it should be $\sigma^2$ not $\sigma$. In general, let's use a different variable besides sigma to refer to the covariance matrix - capital sigma I think could be fine. I think the following sentence should come first and be less conversational. We want to show that each set of assignments to $F, F_{int}, \sigma_\epsilon$ leads to a different distribution of $X_0$ for group $a_0$, i.e. we can uniquely determine the values of these three parameters by observing $P(X_0 \mid A = a_0)$. To do this, we show that the mapping from the parameter values to observed distribution $P(X_0 \mid A = a_0)$ is an injective function — we assume there are two sets of parameters $\{F, F_{int}, \sigma_\epsilon\}$ and $\{F', F_{int}', \sigma_\epsilon'\}$ that lead to the same observed distribution of $X_0$ and show that the parameter values must be equal.
>
> Assuming the two sets of parameters map to distributions of $X_0$ with the same mean, it must hold that $F_{int} = F_{int}'$. Thus, parameter $F_{int}$ is identified by data distribution $P(X_0 \mid A = a_0)$.
>
> Further, the covariance matrix of $X_0$ induced by each set of parameters must be the same: $F(F)^T + \sigma_\epsilon = F'(F')^T + \sigma_\epsilon'$. Element-wise equality of the covariance matrix gives us the following, where subscripts $i$ refer to the $i$-th element of each parameter vector:
>
> $$F_i F_j = F_i' F_j' \;\; \forall i, j, i \neq j \tag{2}$$
>
> $$(F_i)^2 + \sigma_{\epsilon i} = (F_i')^2 + \sigma_{\epsilon i}' \tag{3}$$
>
> Combining equality constraint (2) for multiple pairs of indices, we have that for all assignments of distinct indices $i, j, k$:
>
> $$(F_i F_j = F_i' F_j') \wedge (F_i F_k = F_i' F_k') \implies \frac{F_j'}{F_j} = \frac{F_k'}{F_k}$$
>
> $$(F_j F_k = F_j' F_k') \wedge \left( \frac{F_j'}{F_j} = \frac{F_k'}{F_k} \right) \implies (F_j = \alpha F_j') \wedge (F_k = \alpha F_k'),$$

where $\alpha \in \{-1, +1\}$not exactly sure how second line follows, is there some way to better-epxlain the argument?. Since we have fixed $F_0 > 0$ for all factor loading matrices $F$, we have:

$$F_0 = \alpha F_0' \implies \alpha = 1 \implies F_i = F_i' \;\; \forall i \in [0, d), \tag{4}$$

meaning we have identified $F$.

Lastly, using equations (3) and (4) we get $F_i = F_i' \implies \sigma_{\epsilon i} = \sigma_{\epsilon_i}'$. We have now shown that if two parameter sets induce the same distribution of $X$ at time $t = 0$, they must have the same exact value assignments. Therefore $F, F_{int}, \sigma_\epsilon$ are identified by $P(X_t \mid A = a_0)$. $\qquad\square$

**Lemma A.2.** *Parameters $\mu_{Z_0}^{(a)}, \sigma_{Z_0}^{(a)}, \mu_R^{(a)}, \sigma_R^{(a)}$ are identified by $P(X_t \mid A = a)$ for all groups a*I might write this using the full set of parameters, including F etc (those covered in lemma 1). And I'm not sure I would say for all groups $a$; I might just say $p(X_t|A)$.

*Proof.* Since we have shown that $F, F_{int}, \sigma_\epsilon$ are identified by themselves based on the observed data, we take their values as given in this argumentlet's say this more formally. Ideally I think we should just keep saying throughout "we show that if two parameter sets X and X' yield the same observed data distribution p(blar), they must be identical. By Lemma 1, we know that if subsetX and subsetX' yield same distirbution subsetBlar, they must be identical. [Rest of proof].. For each group $a$, we model the following:

$$Z_0 \sim N\left(\mu_{Z_0}^{(a)}, \sigma_{Z_0}^{(a)}\right)$$

$$R \sim N\left(\mu_R^{(a)}, \sigma_R^{(a)}\right)$$

$$Z_t = Z_0 + R \cdot t \implies Z_t \sim N\left(\mu_R^{(a)} \cdot t + \mu_{Z_0}^{(a)}, \sigma_R^{(a)} \cdot t^2 + \sigma_{Z_0}^{(a)}\right)$$

$$X_t = F \cdot Z_t + F_{int} + \epsilon_t, \text{ where } \epsilon_t \sim N(0, \sigma_\epsilon) \tag{5}$$

lowercase sigma standardly refers to standard deviation, not covariance, so I think some of the entries above should be $\sigma^2$ probably also we should find a notation for the intercept term besides $F_{int}$, which is a bit clunky.one other notational thing - might be easier to use tilde for the alternate parameters not prime - e.g. $\tilde{\mu}^{(a)}$ takes up less space because the tilde just goes over the letter For convenience we will omit the $(a)$ superscript for the rest of the proof. We see that equation (5) captures a factor analysis model with factor loading matrix $F$ and diagonal covariance matrix $\sigma_\epsilon$. So we have that

$$X_t \sim N(F_{int} + F(\mu_R \cdot t + \mu_{Z_0}), F(\sigma_R \cdot t^2 + \sigma_{Z_0})F^T + \sigma_\epsilon).$$

We want to show that every set of assignments to $\mu_{Z_0}, \sigma_{Z_0}, \mu_R, \sigma_R$ leads to a different distribution of $X_t$ at any time $t$, i.e. we can uniquely determine the values of these four parameters by observing $P(X_t \mid A = a)$. To do this, we show that the mapping from the parameter values to observed distribution $P(X_t \mid A = a)$ is an injective function — we assume there are two sets of parameters $\{\mu_{Z_0}, \sigma_{Z_0}, \mu_R, \sigma_R\}$ and $\{\mu_{Z_0}', \sigma_{Z_0}', \mu_R', \sigma_R'\}$ that lead to the same observed distribution of $X_t$ at all $t$.

We first consider $t = 0$, where $X_0 \sim N(F_{int} + F\mu_{Z_0}, F(\sigma_{Z_0})F^T + \sigma_\epsilon)$. For the two parameter sets to map to distributions of $X_0$ with the same mean, it must hold that

$$F_{int} + F\mu_{Z_0} = F_{int} + F\mu_{Z_0}' \implies \mu_{Z_0} = \mu_{Z_0}',$$

and for the two parameter sets to map to distributions with the same covariance matrix, it must hold that

$$F(\sigma_{Z_0})F^T + \sigma_\epsilon = F(\sigma_{Z_0}')F^T + \sigma_\epsilon \implies \sigma_{Z_0} = \sigma_{Z_0}'.$$

So we have identified $\mu_{Z_0}$ and $\sigma_{Z_0}$. We next consider any time $t \neq 0$, where $X_t \sim N(F_{int} + F(\mu_R \cdot t + \mu_{Z_0}), F(\sigma_R \cdot t^2 + \sigma_{Z_0})F^T + \sigma_\epsilon)$. For the two parameter sets to map to distributions of $X_t$ with the same mean, it must hold that

$$F_{int} + F(\mu_R \cdot t + \mu_{Z_0}) = F_{int} + F(\mu_R' \cdot t + \mu_{Z_0}') \implies \mu_R = \mu_R',$$

this looks right, but I might say explicitly it follows because we've already shown that $\mu_{Z0}$ must equal $\mu_{Z0}'$, and similarly below.

and for the two parameter sets to map to distributions with the same covariance matrix, it must hold that

$$F(\sigma_R \cdot t^2 + \sigma_{Z_0})F^T + \sigma_\epsilon = F(\sigma_R' \cdot t^2 + \sigma_{Z_0}')F^T + \sigma_\epsilon \implies \sigma_R = \sigma_R'.$$

So we have identified $\mu_R$ and $\sigma_R$. Thus we have shown that for any group $a$, group-specific values of $\mu_{Z_0}, \sigma_{Z_0}, \mu_R, \sigma_R$ are identified by $P(X_t \mid A = a)$.

$\square$

**Lemma A.3.** *Parameters $\beta_0, \beta_Z$ are identified by $P(D_t \mid Z_t, A = a_0)$this can't be quite the right theorem statement because we dont' observe $Z_t$; I think we want to say $p(D|A, t)$.*

*Proof.* Since we have shown that all group-specific distribution parameters $\mu_R^{(a)}, \sigma_R^{(a)}$ are identified by the observed data, we take their values as given in this argument. This means that we know the distributions of $Z_0$ (pinned) and $R$ for group $a_0$rewrite more formally as suggested above. In addition, we observe each event when a patient in group $a_0$ visits the hospital ($D_t = 1$), which means that the value $\lambda_t$ can be recovered for all timepoints $t$. As described in §3, we model $\lambda_t$ as a function of severity $Z_t$ and demographic group. More specifically, we have $\log(\lambda_t) = \beta_0 + \beta_Z \cdot Z_t + \beta_A^{(a)}$. We define $\beta_A^{(a_0)}$ as 0 for reference, so for group $a_0$ we have $\log(\lambda_t) = \beta_0 + \beta_Z \cdot Z_t = \beta_0 + \beta_Z(Z_0 + R \cdot t)$.

We want to show that our observations of patient visits identify the parameters $\beta_0$ and $\beta_Z$. First, we find it is more straightforward to reason abouttoo informal $\log(\lambda_t)$, which has a one-to-one correspondence with $\lambda_t$ since $\lambda_t$ is positive and $\log(\cdot)$ is a bijection over $\mathbb{R}^+$. Further, instead of the value $\log(\lambda_t)$ itself, which is dependent on each individual patient's value of $Z_0$ and $R$, we reason about the expectation of $\log(\lambda_t)$ over the known group-level distributions of $Z_0$ and $R$. Each set of observations $\mathbb{E}_{Z_0,R}[\log(\lambda_t)] \ \forall t$ *uniquely* defines the visit distribution of the group $a_0$ over time, so by showing that different parameters $\beta_0, \beta_Z$ lead to different values of $\mathbb{E}_{Z_0,R}[\log(\lambda_t)]$ we complete the proof that unique parameters $\beta_0, \beta_Z$ lead to a unique distribution of visit times over group $a_0$I think this is true, but we need to make the argument more succinct + precise. I think you're basically trying to say that if two distributions have unique $E[log(lambda)]$, they must have unique $p(D|t)$. So if we can show that different parameter sets yield unique $E[log(lambda)]$ they must have unique $p(D|t)$. And then we just show that different parameter sets yield unique $E[log(lambda)]$. But we need to make the first part of the claim more precise and actually show it's true. I think one way to do this is to argue that distributions with unique $E[log(\lambda)]$ have unique $E[\lambda]$, and then use the definition of $p(D)$ in terms of lambda to argue taht if you have unique $E[\lambda]$ you have unique $p(D)$.?.

We want to show that every set of assignments $\beta_0, \beta_Z$ leads to a unique observation of $\mathbb{E}_{Z_0,R}[\log(\lambda_t)] = \mathbb{E}_{Z_0,R}[\beta_0 + \beta_Z(Z_0 + R \cdot t)]$ across time $t$. To do this, we show that the mapping from parameter values to the expected value of $\log(\lambda_t)$ over group $a_0$ is an injective function — we assume there are two sets of parameters $\{\beta_0, \beta_Z\}$ and $\{\beta_0', \beta_Z'\}$ that generate the same observed values $\mathbb{E}_{Z_0,R}[\log(\lambda_t)]$ at all timesteps $t$. We want to show it must be the case that $\beta_0 = \beta_0'$ and $\beta_Z = \beta_Z'$.

We first consider some timestep $t'$ such that we observe data at $t = t'$ and $t = t'+1$. At timestep $t'$, we observe:

$$\mathbb{E}_{Z_0,R}[\beta_0 + \beta_Z \cdot Z_0 + \beta_Z \cdot R \cdot t'] = \mathbb{E}_{Z_0,R}[\beta_0' + \beta_Z' \cdot Z_0 + \beta_Z' \cdot R \cdot t']. \quad (6)$$

At timestep $t' + 1$, we observe:

$$\mathbb{E}_{Z_0,R}[\beta_0 + \beta_Z \cdot Z_0 + \beta_Z \cdot R \cdot (t'+1)] = \mathbb{E}_{Z_0,R}[\beta_0' + \beta_Z' \cdot Z_0 + \beta_Z' \cdot R \cdot (t'+1)]. \quad (7)$$

Using linearity of expectation to combine results from (6) and (7), we have that

$$\mathbb{E}_{Z_0,R}[\beta_Z \cdot R] = \mathbb{E}_{Z_0,R}[\beta_Z' \cdot R] \implies \beta_Z \cdot \mathbb{E}_{Z_0,R}[R] = \beta_Z' \cdot \mathbb{E}_{Z_0,R}[R] \implies \beta_Z = \beta_Z'.$$

hm, this doesn't follow if E[R] is 0?

So we have identified $\beta_Z$. We also note that at $t = 0$:

$$\mathbb{E}_{Z_0,R}[\beta_0 + \beta_Z \cdot Z_0] = \mathbb{E}_{Z_0,R}[\beta_0' + \beta_Z' \cdot Z_0]$$
$$\implies \beta_0 + \beta_Z \cdot \mathbb{E}_{Z_0,R}[Z_0] = \beta_0' + \beta_Z' \cdot \mathbb{E}_{Z_0,R}[Z_0]$$
$$\implies \beta_0 = \beta_0'$$

Thus we have shown that $\beta_0, \beta_Z$ are identified by $P(D_t \mid Z_t, A = a_0)$.

$\square$

**Lemma A.4.** *Parameters $\beta_A^{(a)}$ is identified by $P(D_t \mid Z_t, A = a)$ for all other groups $a$.*

*Proof.* I'm willing to believe that similar reasoning works here if it works on the last part, but let's clean up the last part first.Since we have shown that all group-specific distribution parameters $\mu_{Z_0}^{(a)}, \sigma_{Z_0}^{(a)}, \mu_R^{(a)}, \sigma_R^{(a)}$ are identified by the observed data, as well as group-agnostic parameters of the poisson process $\beta_0, \beta_Z$, we take their values as given in this argument. We use an approach very similar to that for Lemma A.3. We let $\mathscr{D}_a$ denote the distributions of $Z_0$ and $R$ for group $a$ (parameterized by $\mu_{Z_0}^{(a)}, \sigma_{Z_0}^{(a)}, \mu_R^{(a)}, \sigma_R^{(a)}$). Then, since each set of observations $\mathbb{E}_{Z_0,R\sim\mathscr{D}_a}[\log(\lambda_t)] \ \forall t$ uniquely characterizes the distribution of visits for group $a$ over time, we can prove identifiability by showing that different values of $\beta_A^{(a)}$ will induce different values of $\mathbb{E}_{Z_0,R\sim\mathscr{D}_a}[\log(\lambda_t)]$. Note that we omit the $(a)$ superscript for the rest of the proof, since we only reason about one group at a time.

We want to show that every value of $\beta_A$ leads to a unique observation of $\mathbb{E}_{Z_0,R\sim\mathscr{D}_a}[\log(\lambda_t)]$ across time $t$. To do this, we show that the mapping from $\beta_A$ to the expected value of $\log(\lambda_t)$ over group $a$ is an injective function — we assume there are two values $\beta_A$ and $\beta_A'$ that generate the same observed values $\mathbb{E}_{Z_0,R\sim\mathscr{D}_a}[\log(\lambda_t)]$ at all timesteps $t$. We want to show it must be the case that $\beta_A = \beta_A'$.

As described in §3, $\log(\lambda_t) = \beta_0 + \beta_Z \cdot Z_t + \beta_A = \beta_0 + \beta_Z(Z_0 + R \cdot t) + \beta_A$. Considering an arbitrary time $t$, we have by assumption that

$$\mathbb{E}_{Z_0,R\sim\mathscr{D}_a}[\beta_0 + \beta_Z(Z_0 + R \cdot t) + \beta_A] = \mathbb{E}_{Z_0,R\sim\mathscr{D}_a}[\beta_0 + \beta_Z(Z_0 + R \cdot t) + \beta_A']$$
$$\implies \beta_0 + \beta_Z \cdot \mathbb{E}_{Z_0,R\sim\mathscr{D}_a}[Z_0 + R \cdot t] + \beta_A = \beta_0 + \beta_Z \cdot \mathbb{E}_{Z_0,R\sim\mathscr{D}_a}[Z_0 + R \cdot t] + \beta_A'$$
$$\implies \beta_A = \beta_A'$$

Thus we have shown that $\beta_A$ is identified by $P(D_t \mid Z_t, A = a)$ for all other groups $a$.

$\square$

By showing that each parameter of the model is uniquely recovered from the observed data, we have proved that our model is identifiable.

$\square$

# B  Proofs of Bias

In this section, we assume that all PDFs and conditional PDFs have positive support over their entire domain. We also assume that all PDFs are differentiable.

## B.1 Proof of Theorem 4.2

We make the following assumptions about the existence of disparities in our setting:

**Assumption B.1.** A patient's severity over time can be estimated by $Z_t = f(R, t) + Z_0$, where $f$ is monotonically increasing in progression rate $R$.

**Assumption B.2.** There exists some underserved group $a$ that tends to start receiving care at later, more severe stages of their disease: $P(Z_0 = z_0 \mid A = a)$ strictly MLRPs $P(Z_0 = z_0)$ with respect to $Z_0$, i.e. $\frac{P(Z_0 = z_0 \mid A = a)}{P(Z_0 = z_0)}$ is a strictly increasing function of $Z_0$.

**Assumption B.3.** On average, this underserved group progresses no slower than the overall population: $\mathbb{E}[R \mid X_t = x_t, A = a] \geq \mathbb{E}[R \mid X_t = x_t]$.

*Proof.* We want to show that $\mathbb{E}[Z_t \mid X_t = x_t, A = a] > \mathbb{E}[Z_t \mid X_t = x_t]$. We first show that $P(Z_0 = z_0 \mid X_t = x, A = a)$ strictly MLRPs $P(Z_0 = z_0 \mid X_t = x_t)$ with respect to $Z_0$:

$$\frac{\partial}{\partial Z_0} \left( \frac{P(Z_0 = z_0 \mid X_t = x_t, A = a)}{P(Z_0 = z_0 \mid X_t = x_t)} \right) = \frac{\partial}{\partial Z_0} \left( \frac{\frac{P(X_t = x_t \mid Z_0 = z_0, A = a) P(Z_0 = z_0 \mid A = a)}{P(X_t = x_t \mid A = a)}}{\frac{P(X_t = x_t \mid Z_0 = z_0) P(Z_0 = z_0)}{P(X_t = x_t)}} \right)$$
(Bayes Rule)

$$= \frac{\partial}{\partial Z_0} \left( \frac{\frac{P(Z_0 = z_0 \mid A = a)}{P(X_t = x_t \mid A = a)}}{\frac{P(Z_0 = z_0)}{P(X_t = x_t)}} \right) \qquad (X_t \perp A \mid Z_0, R)$$

$$= \frac{P(X_t = x_t)}{P(X_t = x_t \mid A = a)} \cdot \frac{\partial}{\partial Z_0} \left( \frac{P(Z_0 = z_0 \mid A = a)}{P(Z_0 = z_0)} \right)$$
$$> 0 \qquad \text{(Assumption B.2)}$$

Since MLRP implies FOSD [43], this also implies that $P(Z_0 = z_0 \mid X_t = x_t, A = a)$ strictly FOSDs $P(Z_0 = z_0 \mid X_t = x_t)$. It follows directly that $\mathbb{E}[Z_0 \mid X_t = x_t, A = a] > \mathbb{E}[Z_0 \mid X_t = x_t]$.

Furthermore,

$$\mathbb{E}[R \mid X_t = x_t, A = a] \geq \mathbb{E}[R \mid X_t = x_t] \qquad \text{(Assumption B.3)}$$
$$\implies \mathbb{E}[f(R, t) \mid X_t = x_t, A = a] \geq \mathbb{E}[f(R, t) \mid X_t = x_t], \quad \forall t \geq 0 \qquad \text{(Assumption B.1)}$$
$$\implies \mathbb{E}[f(R, t) \mid X_t = x_t, A = a] + \mathbb{E}[Z_0 \mid X_t = x_t, A = a]$$
$$> \mathbb{E}[f(R, t) \mid X_t = x_t] + \mathbb{E}[Z_0 \mid X_t = x_t], \quad \forall t \geq 0$$
$$\implies \mathbb{E}[f(R, t) + Z_0 \mid X_t = x_t, A = a] > \mathbb{E}[f(R, t) + Z_0 \mid X_t = x_t], \quad \forall t \geq 0$$
$$\implies \mathbb{E}[Z_t \mid X_t = x_t, A = a] > \mathbb{E}[Z_t \mid X_t = x_t]$$

It is clear to see that this argument extends naturally to show that if a group is "overserved", i.e. they tend to get care earlier than the rest of the population, that their severity will be overestimated: If there exists a group $a'$ such that $P(Z_0 = z_0)$ strictly MLRPs $P(Z_0 = z_0 \mid A = a')$ with respect to $Z_0$ and $\mathbb{E}[R \mid X_t = x_t] \geq \mathbb{E}[R \mid X_t = x_t, A = a']$, then we will see that $\mathbb{E}[Z_t \mid X_t = x_t, A = a'] < \mathbb{E}[Z_t \mid X_t = x_t]$. Hence any model that does not take into account demographic disparities in initial disease severity levels at a patient's first visit will lead to biased estimates of severity. $\square$

## B.2 Proof of Theorem 4.3

We make the following assumptions about the existence of disparities in our setting:

**Assumption B.4.** A patient's severity over time can be estimated by $Z_t = f(R, t) + Z_0$, where $f$ is strictly monotonically increasing in progression rate $R$.

**Assumption B.5.** There exists some group $a$ that tends to progress more quickly: $P(R = r \mid A = a)$ strictly MLRPs $P(R = r)$ with respect to $R$, i.e. $\frac{P(R = r \mid A = a)}{P(R = r)}$ is a strictly increasing function of $R$.

**Assumption B.6.** On average, this underserved group is, on average, first observed no earlier than the overall population: $\mathbb{E}[Z_0 \mid X_t = x_t, A = a] \geq \mathbb{E}[Z_0 \mid X_t = x_t]$.

*Proof.* We want to show that $\mathbb{E}[Z_t \mid X_t = x_t, A = a] > \mathbb{E}[Z_t \mid X_t = x_t]$. We first show that $P(R = r \mid X_t = x_t, A = a)$ strictly MLRPs $P(R = r \mid X_t = x_t)$ with respect to $R$:

$$\frac{\partial}{\partial R}\left(\frac{P(R = r \mid X_t = x_t, A = a)}{P(R = r \mid X_t = x_t)}\right) = \frac{\partial}{\partial R}\left(\frac{\frac{P(X_t = x_t \mid R = r, A = a)P(R = r \mid A = a)}{P(X_t = x_t \mid A = a)}}{\frac{P(X_t = x_t \mid R = r)P(Z_t = z_t)}{P(X_t = x_t)}}\right) \quad \text{(Bayes Rule)}$$

$$= \frac{\partial}{\partial R}\left(\frac{\frac{P(R = r \mid A = a)}{P(X_t = x_t \mid A = a)}}{\frac{P(R = r)}{P(X_t = x_t)}}\right) \quad (X \perp A \mid Z_0, R)$$

$$= \frac{P(X_t = x_t)}{P(X_t = x_t \mid A = a)} \cdot \frac{\partial}{\partial R}\left(\frac{P(R = r \mid A = a)}{P(R = r)}\right)$$

$$> 0 \quad \text{(Assumption B.5)}$$

Since MLRP implies FOSD [43], this also implies that $P(R = r \mid X_t = x_t, A = a)$ strictly FOSDs $P(R = r \mid X_t = x_t)$. It follows directly that:

$$\mathbb{E}[R \mid X_t = x_t, A = a] > \mathbb{E}[R \mid X_t = x_t]$$
$$\implies \mathbb{E}[f(R, t) \mid X_t = x_t, A = a] > \mathbb{E}[f(R, t) \mid X_t = x_t], \quad \forall t > 0 \quad \text{(Assumption B.4)}$$
$$\implies \mathbb{E}[f(R, t) \mid X_t = x_t, A = a] + \mathbb{E}[Z_0 \mid X_t = x_t, A = a]$$
$$> \mathbb{E}[f(R, t) \mid X_t = x_t] + \mathbb{E}[Z_0 \mid X_t = x_t], \quad \forall t > 0 \quad \text{(Assumption B.6)}$$
$$\implies \mathbb{E}[f(R, t) + Z_0 \mid X_t = x_t, A = a] > \mathbb{E}[f(R, t) + Z_0 \mid X_t = x_t], \quad \forall t > 0$$
$$\implies \mathbb{E}[Z_t \mid X_t = x_t, A = a] > \mathbb{E}[Z_t \mid X_t = x_t]$$

It is clear to see that this argument extends naturally to show that if a group is "overserved", i.e. they tend to progress more slowly than the rest of the population, that their severity will be overestimated: if there exists a group $a'$ such that $P(R = r)$ strictly MLRPs $P(R = r \mid A = a')$ with respect to $R$ and $\mathbb{E}[Z_0 \mid X_t = x_t] \geq \mathbb{E}[Z_0 \mid X_t = x_t, A = a']$, then we will see that $\mathbb{E}[Z_t \mid X_t = x_t, A = a'] < \mathbb{E}[Z_t \mid X_t = x_t]$. Thus any model that does not take into account demographic disparities in patient progression rates will lead to biased estimates of severity. $\quad\square$

## B.3 Proof of Theorem 4.4

We make the following assumptions about the existence of disparities in our setting and patient visit rates:

**Assumption B.7.** A patient's visit pattern can be estimated using an inhomogeneous poisson process characterized by visit rate $\Lambda$, such that $\log(\Lambda) = g(Z_t) + \beta_A^{(A)}$ for some function of severity $g(Z_t)$ and group-specific adjustments $\beta_A^{(A)}$.

**Assumption B.8.** There exists some group $a$ that tends to receive care less frequently than other groups, conditional on disease severity: $\beta_A^{(a)} < \beta_A^{(A)}$ for all $A \neq a$.

**Assumption B.9.** Visit rate increases with disease severity: $g(Z_t)$ is a strictly monotonically increasing function of severity.

*Proof.* We want to show that $\mathbb{E}[Z_t \mid \Lambda = \lambda, A = a] > \mathbb{E}[Z_t \mid \Lambda = \lambda]$. We do this by calculating each term separately.

We first consider $\mathbb{E}[Z_t \mid \Lambda = \lambda, A = a]$. The strictly monotone assumption in B.9 ensures $g$ is invertible, and the fact that all visit rates $\Lambda$ are characterized by $\log(\Lambda) = g(Z_t) + \beta_A^{(A)}$ ensures that this holds over the entire range of $\Lambda$ values. This gives us:

$$\mathbb{E}[Z_t \mid \Lambda = \lambda, A = a] = \mathbb{E}\left[g^{-1}\left(\log(\Lambda) - \beta_A^{(A)}\right) \,\Big|\, \Lambda = \lambda, A = a\right]$$

$$= g^{-1}\left(\log(\lambda) - \beta_A^{(a)}\right)$$

We next consider the case where a model infers severity without taking into account disparities in visit rate conditional on severity. Estimating severity $Z_t$ based solely on visit observations gives:

$$\mathbb{E}[Z_t \mid \Lambda = \lambda] = P(A = a) \cdot \mathbb{E}[Z_t \mid \Lambda = \lambda, A = a] + P(A \neq a) \cdot \mathbb{E}[Z_t \mid \Lambda = \lambda, A \neq a]$$

$$= P(A = a) \cdot \mathbb{E}\left[g^{-1}\left(\log(\Lambda) - \beta_A^{(A)}\right) \,\middle|\, \Lambda = \lambda, A = a\right]$$

$$+ P(A \neq a) \cdot \mathbb{E}\left[g^{-1}\left(\log(\Lambda) - \beta_A^{(A)}\right) \,\middle|\, \Lambda = \lambda, A \neq a\right]$$

$$< P(A = a) \cdot \mathbb{E}\left[g^{-1}\left(\log(\Lambda) - \beta_A^{(A)}\right) \,\middle|\, \Lambda = \lambda, A = a\right]$$

$$+ P(A \neq a) \cdot \mathbb{E}\left[g^{-1}\left(\log(\Lambda) - \beta_A^{(a)}\right) \,\middle|\, \Lambda = \lambda, A = a\right] \qquad (*)$$

$$= P(A = a) \cdot \left(g^{-1}\left(\log(\lambda) - \beta_A^{(a)}\right)\right) + P(A \neq a) \cdot \left(g^{-1}\left(\log(\lambda) - \beta_A^{(a)}\right)\right)$$

$$= g^{-1}\left(\log(\lambda) - \beta_A^{(a)}\right)$$

$$= \mathbb{E}[Z_t \mid \Lambda = \lambda, A = a]$$

As justification for $(*)$:

$$\beta_A^{(a)} < \beta_A^{(A)}, \quad \forall A \neq a, \forall \Lambda \qquad \text{(Assumption B.8)}$$

$$\implies \log(\Lambda) - \beta_A^{(a)} > \log(\Lambda) - \beta_A^{(A)}, \quad \forall A \neq a, \forall \Lambda$$

$$\implies g^{-1}\left(\log(\Lambda) - \beta_A^{(a)}\right) > g^{-1}\left(\log(\Lambda) - \beta_A^{(A)}\right), \quad \forall A \neq a, \forall \Lambda$$

$$\text{(Assumption B.9} \implies g^{-1}(Z_t) \text{ strictly monotonically increasing)}$$

$$\implies \mathbb{E}\left[g^{-1}\left(\log(\Lambda) - \beta_A^{(a)}\right) \,\middle|\, \Lambda = \lambda, A = a\right] > \mathbb{E}\left[g^{-1}\left(\log(\Lambda) - \beta_A^{(A)}\right) \,\middle|\, \Lambda = \lambda, A \neq a\right]$$

It is clear to see that this argument extends naturally to show that if a group is "overserved", i.e. they tend to visit the hospital more frequently conditional on severity, that their severity will be overestimated: if there exists a group $a'$ such that $\beta_A^{a')} > \beta_A^{(A)}$ for all $A \neq a'$, then we will see that $\mathbb{E}[Z_t \mid \Lambda = \lambda, A = a'] < \mathbb{E}[Z_t \mid \Lambda = \lambda]$. Thus any model that does not take into account demographic disparities in patient visit rates given their severity will lead to biased estimates of severity. $\qquad \square$

## C   Simulations

Figure 3 shows the results of 50 simulation runs, where we randomly instantiate the parameters of our model and then generate data to fit on. We visualize the recovery of each parameter by plotting true parameter values versus recovered posterior mean values, with one dot per run.

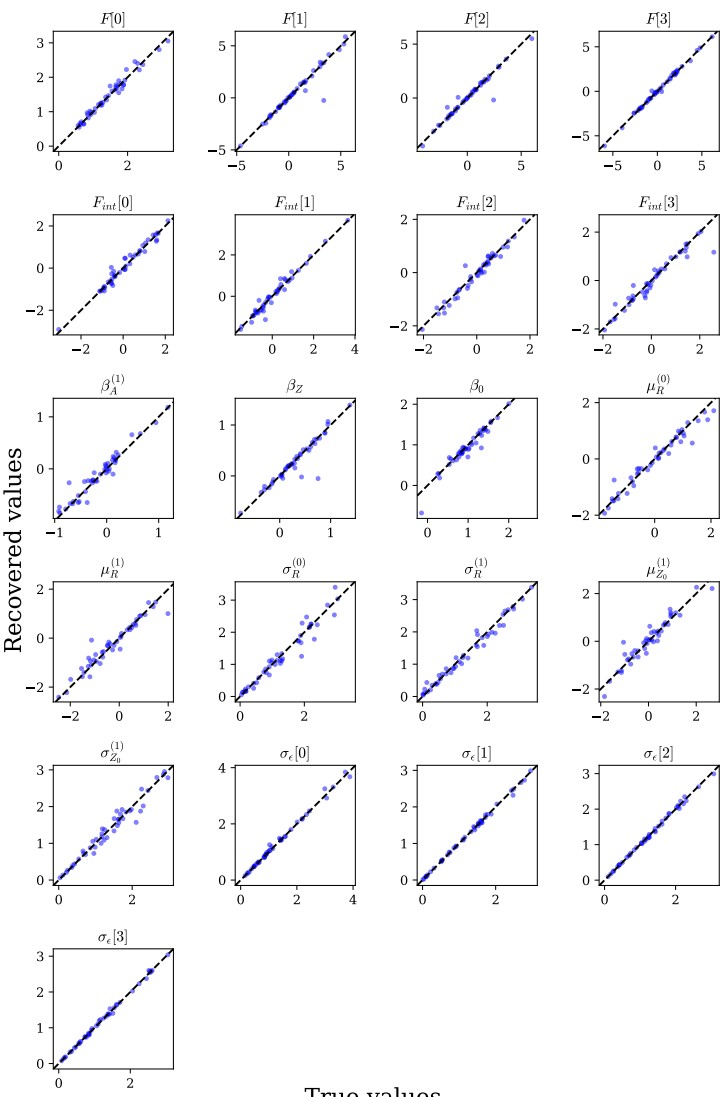

Figure 3: Parameter recovery on 50 runs of fitting our model to synthetic data.

