# OpenReview forum: "Bayesian Disease Progression Models that Capture Health Disparities"
_NeurIPS.cc/2024/Workshop/BDU — Submitted to NeurIPS BDU Workshop 2024_

### Official Review · Reviewer_xsM9 · 2024-09-24
**Review of Bayesian Disease Progression Models that Capture Health Disparities**

**Rating:** 7
**Confidence:** 4

**Review:**

## Summary
This paper presents a Bayesian disease progression model that addresses healthcare disparities, which are typically overlooked in existing models. The authors focus on three specific disparities: (1) certain populations only begin receiving care at more advanced disease stages, (2) some groups experience faster disease progression despite care, and (3) the frequency of care is less for certain populations, even when disease severity is similar. The paper shows both theoretically and empirically that neglecting these disparities leads to biased estimates of disease severity.

## Quality
The quality of the work is high. The authors provide rigorous theoretical analysis and empirical validation using synthetic data.

## Clarity
The paper is generally well-written, with clear motivation and presentation of both the problem and the solution.

## Originality
The paper is original in integrating healthcare disparities into disease progression models.

## Significance
This work is significant due to the growing recognition of healthcare disparities and the need for models that can address them. The potential to collaborate with hospitals for real-world validation enhances the practical relevance of the model. However, the case study results are not yet available, limiting the immediate impact of the work.

## Pros
The model innovatively captures disparities in healthcare, which is a critical and underexplored area. The theorems and proofs support the claims that ignoring disparities biases severity estimates. Additionally, the authors’ collaboration with New York-Presbyterian Hospital for a heart failure case study indicates future real-world impact.

## Cons
The case study is mentioned but not fully explored, which limits the current practical significance. From the model complexity side, while the model is interpretable, a more detailed explanation and comparison with existing models could strengthen the clarity of its contributions.

## Overall Rating
This paper substantially contributes by addressing a critical limitation in disease progression models related to healthcare disparities. It is theoretically sound and has the potential for significant real-world impact.

---

### Official Review · Reviewer_tUMs · 2024-10-09
**Good idea but issues with theory**

**Rating:** 3
**Confidence:** 4

**Review:**

The paper tackles a very interesting problem and proposes an interesting and thoughtful solution. The text in the main body is clearly written.

However, the proof of idenfiability (Theorem 4.1) is incomplete. In particular, it is full of comments that mention changes that should be made and that question the soundness of the proof. Since the main body of the paper claims that the model is identifiable, and currently this claim is not fully supported (beyond simulation), I think that this paper should not be accepted to the workshop. In general, I do not think a paper should be accepted if it contains a theorem with an incomplete or incorrect proof.

Also: the paper is over the page limit. If authors consider resubmission to other venues, I strongly encourage them to follow the page limit.

I think this would be a very strong paper once the issues mentioned above are corrected. I encourage the authors to continue working on this project and submit elsewhere.

---

### Decision · Program_Chairs · 2024-10-09

**Decision:**

Reject

**Comment:**

One of the reviewers raises issues with the paper's theoretical results, in particular due to lack of complete proofs for all claims - in particular, they note that proofs contain comments from coauthors about whether or not they expect arguments to work. As consequence, while the paper is very promising, it is clearly not in a sufficiently-finished state to be ready for presentation.

I encourage the authors to take more time, allow the work to naturally reach a completed state, and resubmit to a future workshop or conference.